# The Unfolded Protein Response: Neutron-Induced Therapy Autophagy as a Promising Treatment Option for Osteosarcoma

**DOI:** 10.3390/ijms21113766

**Published:** 2020-05-26

**Authors:** Ju Yeon Oh, Yeon-Joo Lee, Sei Sai, Tatsuya Ohno, Chang-Bae Kong, Sun Ha Lim, Eun Ho Kim

**Affiliations:** 1Laboratory of Biochemistry, Division of Life Sciences, Korea University, Seongbuk-gu, Seoul 02841, Korea; songoh10@korea.ac.kr; 2Division of Radiation Biomedical Research, Korea Institute of Radiological and Medical Sciences, Seoul 01812, Korea; eyeonjoo@gmail.com; 3Department of Basic Medical Sciences for Radiation Damages, National Institute of Radiological Sciences, National Institutes for Quantum and Radiological Science and Technology, Chiba 263–8555, Japan; sai.sei@qst.go.jp; 4Gunma University Heavy Ion Medical Center, 3–39–22 Showa-machi, Maebashi 371–8511, Japan; tohno@gunma-u.ac.jp; 5Department of Orthopedic Surgery, Korea Institute of Radiological and Medical Sciences, Seoul 139–706, Korea; cbkongmd@gmail.com; 6Department of Biochemistry, School of Medicine, Daegu Catholic University, Duryugongwon-ro, Nam-gu, Daegu 42472, Korea; sunha112@cu.ac.kr

**Keywords:** autophagy, linear energy transfer, neutron beam, osteosarcoma, radiosensitivity, unfolded protein response

## Abstract

Radiotherapy using high linear energy transfer (LET) radiation results in effectively killing tumor cells while minimizing dose (biological effective) to normal tissues to block toxicity. It is well known that high LET radiation leads to lower cell survival per absorbed dose than low LET radiation. High-linear energy transfer (LET) neutron treatment induces autophagy in tumor cells, but its precise mechanisms in osteosarcoma are unknown. Here, we investigated this mechanism and the underlying signaling pathways. Autophagy induction was examined in gamma-ray-treated KHOS/NP and MG63 osteosarcoma cells along with exposure to high-LET neutrons. The relationship between radiosensitivity and autophagy was assessed by plotting the cell surviving fractions against autophagy levels. Neutron treatment increased autophagy rates in irradiated KHOS/NP and MG63 cells; neutrons with high-LETs showed more effective inhibition than those with lower LET gamma-rays. To determine whether the unfolded protein response and Akt-mTOR pathways triggered autophagy, phosphorylated eIF2α and JNK levels, and phospho-Akt, phosphor-mTOR, and phospho-p70S6 levels were, respectively, investigated. High-LET neutron exposure inhibited Akt phosphorylation and increased Beclin 1 expression during the unfolded protein response, thereby enhancing autophagy. The therapeutic efficacy of high-LET neutron radiation was also assessed in vivo using an orthotopic mouse model. Neutron-irradiated mice showed reduced tumor growth without toxicity relative to gamma-ray-treated mice. The effect of high-LET neutron exposure on the expression of signaling proteins LC3, p-elF2a, and p-JNK was investigated by immunohistochemistry. Tumors in high-LET-neutron radiation-treated mice showed higher apoptosis rates, and neutron exposure significantly elevated LC3 expression, and increased p-elF2a and p-JNK expression levels. Overall, these results demonstrate that autophagy is important in radiosensitivity, cell survival, and cellular resistance against high-LET neutron radiation. This correlation between cellular radiosensitivity and autophagy may be used to predict radiosensitivity in osteosarcoma.

## 1. Introduction

Fast neutron therapy (FNT) has been an available cancer radiation treatment since 1976, which is now routinely performed. FNT and boron neutron capture therapy utilize the effects of secondary protons and alpha particles, respectively, which have 5–50-fold higher linear energy transfer (LET) than radiation from electron and proton accelerators typically used in hospitals. High-LET radiation differs from low-LET radiation such as photons, electrons, and high-energy (>100 MeV) protons with respect to their high ionization density, effects on the oxygen concentration in cells, and dependence on the cell cycle phase and on cell differentiation. In particular, high-LET radiation can sustainably deactivate radioresistant tumor cells. However, this property yields a slight therapeutic gap with an increased risk of off-target effects on the normal tissue. The most destructive effect of radiation is DNA damage, which blocks cell proliferation.

High-LET forms of radiation such as fast neutrons lead to tissue damage mainly through nuclear interactions, whereas low-LET forms of radiation cause damage via activated radicals produced by atomic interactions.

Oxygen plays a critical role in free radical production in the body, which is deprived of an adequate oxygen supply under specific pathological hypoxic conditions. Since the tumor microenvironment is hypoxic [1,2], the probability of free radical production in cells of a malignant tumor is decreased. Furthermore, hypoxic cells irradiated with low-LET radiation are radioresistant, whereas high-LET neutrons do not require the presence of oxygen to damage cancer cells [1,3]. Soft-tissue sarcomas are resistant to photon beams; therefore, treatment of these tumors [4,5], as well as melanomas [6,7] and brain tumors [8,9], with neutrons has yielded superior clinical results compared to treatment with photons. Moreover, conventional radiation therapies such as photon treatment have generally been used to control bone sarcomas because of the risk of radiation-induced osteoradionecrosis. In these cases, the absorbed dose is decreased to approximately 25% or less by using low-neutron “kerma” radiation [10]. Therefore, the treatment of bone and cartilage tumors is an important application of clinical neutron therapy [11,12].

Numerous studies have been performed to elucidate the molecular mechanisms by which high-LET radiation induces cell death in cancer cells. Recent reports have shown that cells induce autophagy in response to severe microenvironments such as starvation, hypoxia, and increased reactive oxygen species levels. Autophagy is an evolutionarily conserved process by which cells recycle their components such as long-lived proteins and damaged organelles [13,14,15]. Previous studies reported that high-LET carbon-ion-induced autophagy occurs in cancer cells [16,17,18]. For example, Jin et al. [18]. found that carbon ions efficiently caused autophagy flux in cancer cells, and that the autophagic level increased in a LET-dependent manner. However, the corresponding effects or mechanisms underlying the effectiveness of neutron beam therapy on osteosarcoma (OS) are poorly understood. In this study, we investigated the dependence of autophagy elicited by neutron beam therapy on the LET of radiation in primary cells from patients with OS in vitro and in a mouse orthotopic model in vivo. Furthermore, the molecular mechanisms and signaling pathways regulating the autophagic response to high-LET neutrons were explored. 

## 2. Results

### 2.1. Neutrons Induce Autophagy in a Dose-Dependent Manner in OS Cells 

Figure 1a shows the autophagy levels in KHOS/NP and MG63 cells irradiated with gamma-ray and LET neutrons of different levels at 48 h post-irradiation. Neutron radiation enhanced irradiation-induced autophagy more effectively than gamma-ray radiation, and the apoptotic rate of KHOS/NP and MG63 cells increased at the indicated time points after irradiation in neutron-treated cells compared to that in gamma-ray-treated cells (Figure 1b). Autophagy was quantified using the Cyto-ID reagent, which specifically fluoresces in autophagic vesicles. As shown in Figure 1c, an increased level of the Cyto-ID green autophagy dye accumulated around neutron-treated KHOS/NP, MG63 cells, and primary OS cells derived from a patient with OS. Conjugation of the soluble form of LC3 (LC3-I) with phosphatidylethanolamine and its conversion into a nonsoluble form (LC3-II) are hallmarks of autophagy [19]; thus, we further examined the expression of LC3-II in the absence or presence of chloroquine (CQ). After irradiation at the indicated time points, LC3-II levels were increased in the two cell lines (Figure 1d). In addition, the Neutron increased the expression of LC3-II despite CQ. Notably, the expression of LC3-II following neutron irradiation was slightly higher than that observed after gamma-ray irradiation. 

Neutron treatment significantly inhibited cell growth, as determined by a trypan blue-exclusion assay in the two OS cell lines and primary cells derived from a patient with OS at each time point (Figure 1e). Collectively, these data confirmed that irradiation-induced autophagy in KHOS/NP and MG63 cells and the autophagic cell death levels of irradiated cells increased in response to neutron treatment compared to those treated with gamma-rays. As shown in Figure 1F, FACS analysis revealed that neutrons delayed the S and G2/M phase and induced cell death in OS cell lines.

### 2.2. Relationship Between Autophagy and Cell Survival After Exposure to Gamma-Rays and LET Neutrons

Next, to select the appropriate conditions, we treated OS cells with gamma-rays and neutrons and examined the staining of Cyto-ID reagent (Figure 2a). Ultrastructural analysis showed that compared to the control group, Giemsa-stained KHOS/NP and MG63 cells treated with neutrons for up to 48 h exhibited morphological changes throughout the cytoplasm and membrane, including loss of plasma membrane integrity and obvious vacuole formation; these data are consistent with the morphological changes observed by Giemsa staining (Figure 2b). In addition, transmission electron microscopy was performed to verify the formation of autophagosomes in neutron-treated cells. As shown in Figure 2c, neutron-treated cells exhibited accumulation of large autophagic vacuoles with a typical double-layer membrane and organelle remnants, whereas only a few vacuoles were observed in the gamma-ray-treated group. The surviving fraction correlated well with the level of autophagy at each radiation dose. A close relationship between survival and autophagy levels in response to radiation was observed, indicating that autophagy plays a role in cellular radiosensitivity (Figure 2d).

To test whether autophagy by neutrons promotes cell death and contributes to cellular sensitivity, we pretreated LY294002 for 24 h, which is a PI3K inhibitor, and evaluated cell death by neutrons. LY294002 rescued reduced cell proliferation by gamma-ray radiation and neutrons. However, despite the effect of LY294002, neutron effectively suppressed cell proliferation in OS cells (Figure 2e,f). Our data indicate that autophagy plays a critical role in radiosensitivity, promotes cell death, and contributes to cellular sensitivity against high-LET radiation.

### 2.3. Neutron Therapy Induces Autophagy in Tumor Cells by Inhibiting the Akt-mTOR Pathway 

To explore the molecular mechanisms underlying the changes in the levels of autophagy induced by gamma-rays or neutrons in OS cells, the Akt-mTOR pathway was examined in KHOS/NP and MG63 cells. mTOR acts as a central regulator of autophagy induction, and Akt modulates the activation of mTOR [15]. Therefore, we first explored whether radiation-induced autophagy occurred via mTOR inhibition. The levels of phospho-Akt (p-Akt), phosphor-mTOR (p-mTOR), and phospho-p70S6 (p-p70S6), all of which are mTOR substrates, were measured in the two tumor cell lines. At 48 h post-irradiation, high-LET neutron exposure caused an obvious decrease in the level of p-Akt proteins in the two cell lines compared with that induced following exposure to gamma-rays. As shown in Figure 3a, high-LET neutron exposure further decreased activation of the Akt-mTOR pathway in OS cells. Moreover, exposure to neutrons resulted in larger reductions in p-Akt, p-mTOR, and p-p70S6 expression levels in the two OS cell lines compared with those observed after exposure to gamma-rays. As shown in Figure 3b, expression levels of the mTOR target proteins p-p70S6K, p-mTOR, and p-Akt were highly reduced in neutron-treated cells, as determined by ELISA. These results indicate that neutron radiation induces autophagy in tumor cells by decreasing activation of the Akt-mTOR pathway. Additionally, these findings suggest that this pathway is more effectively inhibited by exposure to neutrons with high-LETs than to those with relatively low LETs.

### 2.4. Neutrons Induce Autophagy Via Activation of Endoplasmic Reticulum (ER) Stress 

ER stress is one of the signaling pathways involved in regulating autophagy [16]; therefore, we hypothesized that ER stress plays an important role in radiation-induced autophagy. Figure 4a shows the expression levels of Bip, a major indicator of the unfolded protein response (UPR) [17], at 48 h post-irradiation in KHOS/NP and MG63 cells, respectively. The expression of Bip was promoted by radiation, indicating ER stress in the tumor cells after irradiation. eIF2α and JNK are known to be involved in autophagy induction through the UPR [18,19]. The phosphorylated levels of these two proteins were also increased in irradiated cells, as determined by ELISA (Figure 4b), suggesting that neutron radiation-activated UPR is involved in autophagy induction in the two OS cell lines. Next, we examined whether inhibition of UPR using PBA influenced the level of autophagy in the two OS cells exposed to neutrons. In mock-treated cells, PBA did not change the levels of p-eIF2α, p-JNK, but slightly increased the expression levels of beclin1, CHOP, and LC3-II (Figure 4c). However, PBA rescued the neutron-induced UPR, as indicated by decreased Bip and CHOP expression levels and reduced phosphorylation of JNK compared with those observed after irradiation with neutrons alone. These findings indicate that exposure to high-LET neutrons effectively increased Beclin 1 expression under UPR, resulting in enhanced autophagy.

### 2.5. Autophagic Effects of Low-LET Gamma-Ray or High-LET Neutron Exposure on Orthotopic Tumors In Vivo

We next assessed the therapeutic efficacy of high-LET neutron radiation in vivo using an orthotopic mouse model. Neutron irradiation decreased tumor growth in mice compared with that in gamma-ray-treated mice (Figure 5a), with no visible signs of toxicity as evidenced by the lack of difference in body weight (data not shown). Additionally, p-elF2, p-JNK, and LC3 staining revealed that tumors in high-LET neutron-treated mice showed a higher apoptosis rate than those from mice treated with low-LET gamma-ray levels (Figure 5b). Next, we examined the association between high-LET radiation and autophagy in vivo. As shown in Figure 5c, immunohistochemistry demonstrated that exposure to high-LET neutrons resulted in significantly elevated LC3, p-elF2a, and p-JNK expression levels compared to those in the gamma-ray treated group. An overview of the molecular changes induced by high-LET neutron treatment in OS cells and tumors is shown in Figure 5d.

## 3. Discussion

Heavy-ion radiotherapy has recently been suggested as a promising new anticancer strategy, and the molecular mechanisms regarding its anticancer action have been previously investigated. These studies have shown that autophagic levels in tumor cells increase in a LET-dependent manner following carbon ion beam exposure [18]. This is the first study to report that exposure to neutrons influences autophagic levels in tumor cells at the molecular level. In our study, low-LET gamma-rays elicited a slight increase in LC3-II protein expression levels, whereas LC3-II expression after neutron exposure was highly increased in KHOS/NP and MG63 cells. This result was verified by flow cytometry measurements of AVO. Consistent with previous studies [20,21], we observed that high-LET neutron exposure had a weaker effect on cell survival than low-LET gamma-rays. We further analyzed the relationship between the cell count and cell autophagic level after irradiation with low-LET gamma-rays and high-LET neutrons in KHOS/NP and MG63 cells. The increase in the surviving cell number was inversely proportional to the autophagic level. These findings reveal a correlation between cellular radiosensitivity and autophagy, which may be used to predict radiosensitivity in OS exposed to radiation of different qualities. Given that these results, which were obtained by autophagy-related experiments, are consistent with those of previous studies in which autophagy inhibition was shown to increase cellular sensitivity to various therapies [22,23,24], the relationship between the two events should be investigated using rapamycin or chloroquine. Additionally, further analysis is needed to explore the reason underlying the more severe autophagy induction induced by high-LET neutron beams than low-LET gamma-rays. 

To explore the mechanism, we examined the influence of the two types of irradiation on the Akt-mTOR and UPR pathways. mTOR effectively represses autophagy by interacting with the ULK1 kinase complex, and directly phosphorylates the ATG13L and ULK1 subunits to suppress ULK1 kinase activity [25,26]. The present results suggest that neutron exposure effectively inhibited the Akt-mTOR pathway and decreased the expression of p-mTOR. These results are similar to those of Nakagawa et al. [27], who reported that carbon ion beam irradiation suppressed the expression of mTOR and associated proteins compared to low-LET radiation. Furthermore, ER stress has been reported to induce autophagy. Various physiological and pathological conditions may cause the accumulation of unfolded or misfolded proteins, resulting in ER stress, which triggers the UPR as an adaptive response to ensure cell survival or to induce cell death under severe stress conditions. Zhang et al. [28,29] found that X-ray exposure induced upregulation of ER stress markers, including Bip and GRP94, at the protein and mRNA levels in IEC-6 cells. Chiu et al. [29]. showed that increases in IRE1 levels and the phosphorylation of eIF2α after exposure to X-rays were correlated with the degree of DNA damage. Consistent with these previous studies, we found that the expression of Bip, which is a sensor of ER stress [30], was enhanced after irradiation, suggesting that both gamma-rays and high-LET neutron radiation can elicit this effect in OS cells. Furthermore, the Bip expression level was increased in response to neutron beam exposure, indicating that high-LET neutrons caused more severe ER stress than low-LET gamma-rays. In response to ER stress, Bip dissociates from the luminal domains of PERK and activates PERK. Activated PERK phosphorylates eIF2α and mediates autophagy via the ATF4-DDIT3/CHOP-TRIB3-Akt-mTOR axis [31,32]. In addition, Bip dissociation activates IRE1, which then recruits the TNFR-associated factor 2 (TRAF2) to form the IRE1–TRAF2–ASK1 complex that, in turn, phosphorylates JNK [33]. Subsequently, activated JNK phosphorylates Bcl-2 located in the ER, resulting in dissociation of Bcl-2 from Beclin 1 to trigger autophagy [34]. In the present work, the expression levels of key proteins of these signaling pathways were examined to evaluate the phosphorylation of eIF2α and JNK in neutron-irradiated cells, and the results were consistent with those of previous studies.

Autophagy strongly affects cell proliferation and survival or autophagic cell death. Therefore, various trials have been conducted to test the potential of modulating autophagy for improving cancer treatment outcomes in combination with currently used treatment modalities such as radiotherapy in different cancer types. The strategies used can be divided into three main categories. The most common strategy is the induction of autophagy by activating the PI3K–Akt–mTOR pathway, mainly by using mTOR inhibitors. This strategy has been adopted to treat glioblastoma, oral, lung, breast, esophageal, and prostate cancers. The second strategy involves the induction of autophagy through the promotion of the mitogen-activated protein kinase pathway by inducing ER stress. This has been employed in pancreatic, colorectal, and prostate cancers. The third is the inhibition of autophagy through the UPR pathway, which is achieved by the addition of chloroquine, as demonstrated in studies of glioblastoma and colorectal cancer. These three strategies are aimed at modulating the autophagy levels of tumor cells to increase the efficacy of radiotherapy. Based on our results, we propose that Akt-mTOR is located downstream of the UPR pathway and that high-LET carbon ions activate the UPR. The latter induces the phosphorylation of eIF2α and CHOP, which consequently inhibits Akt and mTOR activation by blocking Akt phosphorylation, leading to autophagy. Based on the present findings, we propose a molecular mechanism of autophagy induced by high-LET radiation, which is summarized in Figure 5. 

## 4. Materials and Methods 

### 4.1. Cell Culture and Tissue Samples

The KHOS/NP and MG63 OS cell lines were purchased from the American Type Culture Collection (Manassas, VA, USA). Both OS cell lines were cultured in Dulbecco’s modified Eagle medium (WelGene, Daegu, Korea) containing 10% (*v/v*) fetal bovine serum (Gibco, Grand Island, NY, USA) and 1% (*v/v*) penicillin–streptomycin (Gibco). OS tissue was obtained with informed consent from a patient who underwent surgery at the Korea Institute of Radiological and Medical Sciences (Institutional Review Board Approval Number K-1603-001-001, 2016/3/2), and a primary cell culture was established from this tissue as previously described [35]. The cells were treated with gamma-ray and high-LET neutrons. In addition, cells were treated with phenylbutyric acid (PBA; Sigma-Aldrich, St. Louis, MO, USA) to examine the impact on the unfolded protein response (UPR), as PBA is a chemical chaperone in the endoplasmic reticulum (ER) to prevent activation of the UPR and inhibit ER stress [20].

### 4.2. Irradiation 

Cells were cultured in 60- or 100-mm dishes until 70–80% confluence at 37 °C in a humidified atmosphere of 5% CO_2_. Irradiations were performed using a 137 Cs gamma-ray source (Atomic Energy of Canada, Ltd., Ontario, Canada) at a dose rate of 3.81 Gy/min. Fast neutrons (9.8 MeV, 30–40 keV/µm) were produced by the bombardment of the proton on beryllium 9Be(p,n)10B as a nuclear reaction in the cyclotron (MC-50; Scanditronix, Uppsala, Sweden). Paired ionization chambers were used to measure the absorbed dose and dose distribution of fast neutron beams or gamma-rays [36]. Dosimetric measurements were done before in vitro study to calculate neutron dose using RBE, 2.2, which has been used for neutron therapy in our institute and it showed cell killing efficacy equivalent to that of gamma-ray as determined by clonogenic assay [37]. 

### 4.3. Quantification of Acidic Vacuoles (AVOs) by Acridine Orange (AO) Staining

Autophagy induction was detected based on the presence of AVOs. Cells were treated with this concentration of irradiation for the indicated time points, followed by staining with 1 μM AO for 15 min. The cells were then washed, resuspended in PBS, and subjected to FACS analysis. The green (510–530 nm, FL-1) and red (650 nm, FL-3) fluorescence of AO, following blue (488 nm) excitation, was determined for 10,000 events and measured on a FACScan cytofluorimeter using Cell Quest software (BD Biosciences, Franklin Lakes, NJ, USA).

### 4.4. ELISA 

The activities of p70S6K, mTOR, Akt, JNK, and eIF2a were measured using ELISA kits (Cell Signaling Technology, Danvers, MA, USA) according to the manufacturer’s recommendations. Data were collected using a Multiskan EX reader (Thermo Fisher Scientific, Waltham, MA, USA) at 450 nm [38].

### 4.5. Colony Formation Assay

The cells were treated with gamma-ray and high-LET neutrons for 48 h and then incubated for 7 d. The resulting colonies were fixed with 100% methanol for 30 min, stained with 0.4% crystal violet (Sigma, St. Louis, MO, USA), and then counted. 

### 4.6. Detection of Apoptotic Cells by Annexin V Staining

After the cells were treated with gamma-ray and high-LET neutron radiation, they were incubated for an additional 72 h. The cells were washed with PBS, trypsinized, and resuspended in 1× binding buffer (10 mM HEPES/NaOH (pH 7.4), 140 mM NaCl, and 2.5 mM CaCl_2_) at a density of 1 × 10^6^ cells/mL. Aliquots (100 μL) of the cell suspension were mixed with 5 μL annexin V-fluorescein isothiocyanate (PharMingen, San Diego, CA, USA) and 10 μL propidium iodide stock solution (50 μg/mL in PBS) by gentle vortexing, followed by a 15-min incubation at room temperature in the dark. Buffer (400 μL, 1×) was added to each sample and analyzed on a FACScan flow cytometer (BD Biosciences). A minimum of 10,000 cells were counted for each sample, and data analysis was performed using CellQuest software (BD Biosciences).

### 4.7. Western Blotting

After the cells were treated with gamma-rays and neutrons, they were incubated for 48 h. The cells were lysed with a cell lysis buffer (Cell Signaling Technology). Proteins were separated by sodium-polyacrylamide gel electrophoresis and transferred to nitrocellulose membranes. The membranes were blocked with 1% (*v/v*) nonfat dried milk in Tris-buffered saline with 0.05% Tween-20 and incubated with the following primary antibodies: Beclin 1, LC3 (A/B), p-mTOR (Ser2448), p-Akt (S473), p-P70S6K (Thr389), p-eIF2a (Ser51), P-SAPK/JNK (Thr183/Tyr185), CHOP, Bip, and cleaved PARP (1:1000 dilutions, all purchased from Cell Signaling Technology (Danvers, MA, USA), and antibodies against beta-actin (Santa Cruz Biotechnology, Dallas, TX, USA) as reference. The membranes were then incubated with the secondary antibodies at a 1:5000 dilution. Immunoreactive protein bands were visualized by enhanced chemiluminescence (Thermo Fisher Scientific, Waltham, MA, USA) and scanned.

### 4.8. Morphology 

To examine the effect of irradiation on cell morphology, the irradiated cells were stained with Giemsa. The cells were seeded in six-well plates and allowed to adhere overnight onto cover slips followed by irradiation treatment. The cells were then fixed with methanol for 10 min and stained with Giemsa (10% in PBS) for 15 min, followed by washing with tap water. Images were acquired using a Nikon Eclipse Ts2R-FL microscope (Tokyo, Japan).

### 4.9. Orthotopic Model and Histological Analysis

Twelve 4-week-old female BALB/c nude mice (average weight: 12.1 g, range: 11.3–13.1 g) were purchased from ORIENT Bio (Seoul, Korea) and quarantined for 1 week prior to experimentation. KHOS/NP orthotopic tumors were established as previously described Korea Institute of Radiological and Medical Sciences (Institutional Review Board Approval Number K-1603-001-001, 2016/3/2) [39].

### 4.10. Immunohistochemical Staining

For immunohistochemical assessments, 4-μm-thick paraffin-embedded OS sections were mounted onto coated glass slides to detect the proteins under investigation. Following antigen retrieval and blocking of endogenous peroxidases and nonspecific protein binding, the slide sections were incubated first with primary antibodies against p-eIF2a, P-JNK, LC3 (diluted 1:200) purchased from Cell Signaling Technology, followed by appropriate horseradish peroxidase-conjugated secondary antibodies. 

## 5. Conclusions

In conclusion, the present study demonstrated that high-LET radiation can induce autophagy. Studies of the effect of autophagy in radiosensitizing tumors for improving radiotherapy efficiency have shown controversial results. Therefore, further investigation is necessary before the modulation of autophagy can be applied as a therapeutic strategy in clinical settings. However, this remains challenging because autophagy is difficult to quantify; for example, the number of autophagosomes is not directly related to the level of autophagy, as an increased number of autophagosomes may represent a decreased rate of turnover apart from an increased rate, and a reduced number of autophagosomes may indicate an increase in autophagic flux [40]. In addition, personalized treatments aimed at modifying the tumor microenvironment and treatment modalities warrant further investigation. Particularly, hypoxia and the varied radiotherapy dose scheme, as well as differences in autophagic pathways between preclinical and clinical studies, must be investigated, as cellular pathways involve multiple feedback loops and backup mechanisms. These studies should aim to comprehensively elucidate the mechanisms underlying autophagy and its potential application for cancer treatment to enable modalities such as radiotherapy to be used with higher precision.

## Figures and Tables

**Figure 1 ijms-21-03766-f001:**
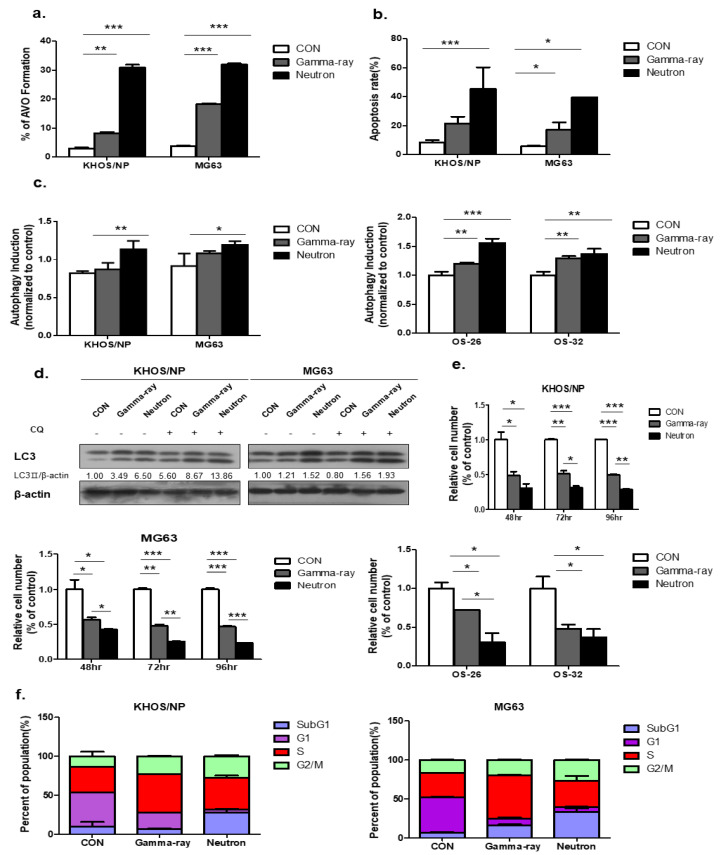
Autophagy induced by neutron exposure. (**a**) Cells were treated with gamma-rays and neutrons for 48 h and then stained with acridine orange (AO). Green and red fluorescence in AO-stained cells were detected by flow cytometry; * *p* < 0.05, ** *p* < 0.01, *** *p* < 0.001. (**b**) Apoptosis rate assessed by FACS analysis after 72 h irradiation (IR) treatment; * *p* < 0.05, *** *p* < 0.001. (**c**) Neutron treatment resulted in an increased Cyto-ID dye signal in KHOS/NP and MG63 cells and primary cells derived from a tumor sample from a patient with osteosarcoma (OS) at 48 h; * *p* < 0.05, ** *p* < 0.01, *** *p* < 0.001. (**d**) Immunoblot of LC3 in KHOS/NP and MG63 cells exposed to neutron radiation for 48 h. (**e**) KHOS/NP and MG63 cells and primary cells from a tumor sample from a patient with OS were treated with each type of radiation for 48 h, and the proliferation rate was measured by cell counting; * *p* < 0.05, ** *p* < 0.01, *** *p* < 0.001. (**f**) After 48 h, the cell cycle distribution was analyzed quantitatively. Values represent the means of three experiments _ SE.

**Figure 2 ijms-21-03766-f002:**
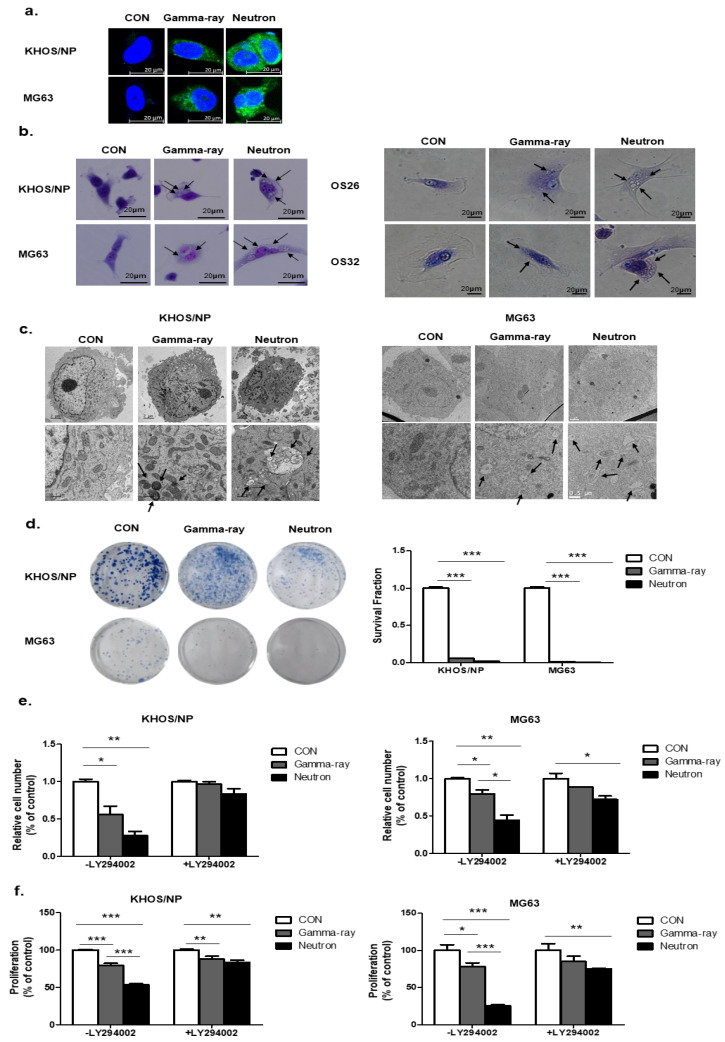
Relationship between autophagy and cell survival after exposure to gamma-rays and neutrons. (**a**) Cyto-ID staining with and without radiation treatment in the KHOS/NP and MG63 cells. (**b**) Cells were stained with Giemsa (10% in PBS), washed, and imaged under a Nikon Eclipse Ts2R-FL microscope (magnification, 40×). Black arrows show vacuoles. An image representative of two independent experiments is shown. (**c**) Autophagy observed by transmission electron microscopy in irradiated OS cells. (**d**) Survival curves of the pharmacological promotion of autophagy on the sensitivity of OS cells to high-LET (high linear energy transfer) radiation; *** *p* < 0.001. The proliferation of OS cell lines were estimated by trypan blue assay (**e**) and MTT (**f**); * *p* < 0.05, ** *p* < 0.01, *** *p* < 0.001.

**Figure 3 ijms-21-03766-f003:**
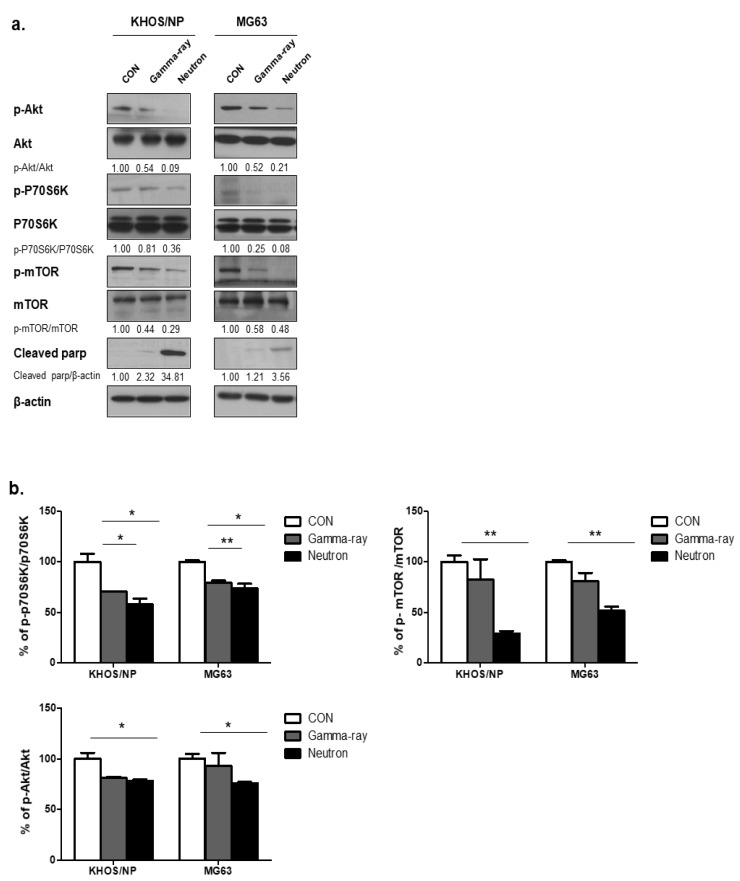
Akt-mTOR signaling pathway activation by gamma-rays or neutrons. (**a**) Immunoblot analysis of p-AKT, p-70S6K, p-mTOR, and cleaved PARP in KHOS/NP and MG63 cells treated with radiation for 48 h. (**b**) ELISA was performed to quantify the levels of phospho-mTOR, mTOR, phospho-AKT, and AKT in KHOS/NP and MG63 cells after irradiation. Each concentration was tested in quadruplicate, and each experiment was repeated twice. The data shown represent the mean ± SD from two independent experiments combined; * *p* < 0.05, ** *p* < 0.01.

**Figure 4 ijms-21-03766-f004:**
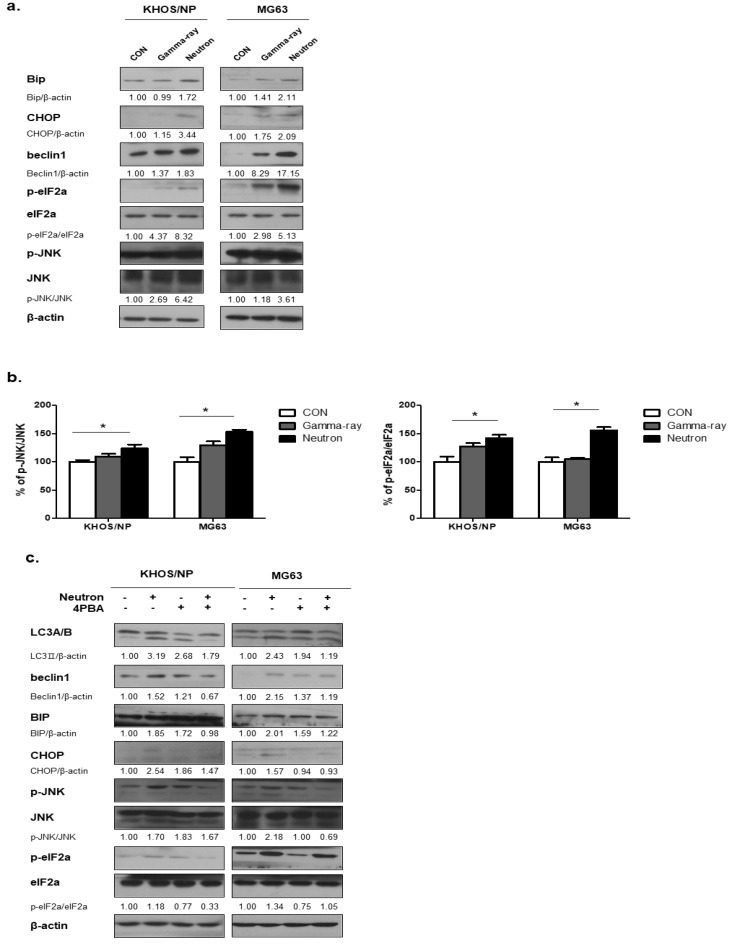
Radiation-induced autophagy via the unfolded protein response. (**a**) Bip and kesy molecules in autophagy were upregulated in response to neutron beam therapy in KHOS/NP and MG63 cells. (**b**) ELISA was performed to quantify the level of phospho-JNK and JNK and phospho-elF2a and elF2a in KHOS/NP and MG63 cells after irradiation. Each concentration was tested in quadruplicate, and each experiment was repeated three times. The data shown represent the mean ± SD from three independent experiments combined; * *p* < 0.05. (**c**) Cell lysates were immunoblotted with the indicated antibodies. PBA treatment prevented activation of the unfolded protein response (UPR) and inhibited autophagy induced by neutrons in OS cells.

**Figure 5 ijms-21-03766-f005:**
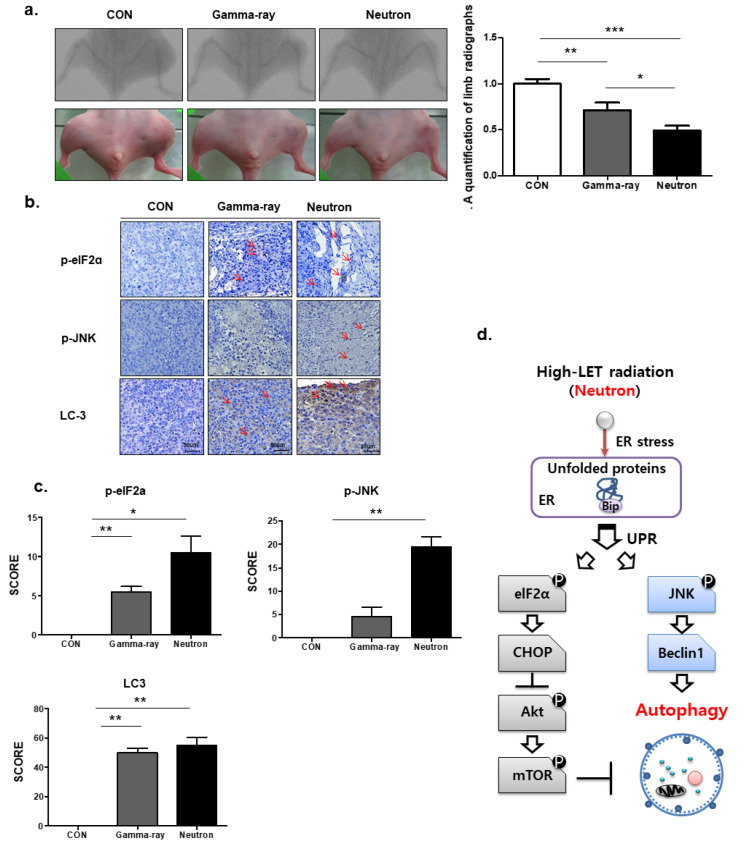
Autophagic effects of neutron beams on orthotopic tumors in vivo. (**a**) KHOS/NP cells were injected into the proximal tibia of four groups each containing three nude mice to generate an orthotopic tumor model. The dimensions of the leg (including the tumor) were measured every 7 d by X-ray analysis. Representative radiographs of the limb of a mouse at 0 and 6 weeks after tumor inoculation are shown. Representative images of animal tumors at 6 weeks and a graph of tumor size against time are shown. (**b**,**c**) p-elF2, p-JNK, and LC3 expression in xenografts was examined by immunohistochemistry. Representative images are shown. Values represent the means ± SD (n = 3); * *p* < 0.05, ** *p* < 0.01. (**d**) The proposed signaling pathways involved in the autophagic cell death by neutrons in OS cells.

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
