# Peer review of "The Unfolded Protein Response: Neutron-Induced Therapy Autophagy as a Promising Treatment Option for Osteosarcoma"

_ijms, 2020, doi:10.3390/ijms21113766_

Round 1
Reviewer 1 Report
The article by Oh et al. explores the role of autophagy induced by high-linear energy transfer neutron in osteosarcoma models. The manuscript is well written, the aims and objectives are focused. However, many conclusions are not sustained by the experiments carried. Experimental plain, indeed, is sequential but the interpretation of the results appears to be very personal.
The supporting bibliography is ancient. From an analysis on PubMed it is possible to refer to much more recent articles, also leaving the works that are milestones of the topic.
Major comments
To sustain the conclusions about the role exerted by autophagy in the experimental model, it is necessary to carry out some experiments in which autophagy is inhibited either by specific chemical inhibitors or by silencing some protein markers. This will allow to better understand the role of the autophagic pathway in the experimental model studied.
Western blotting analyses do not report densitometric analysis. Therefore, it is not easy to analyse the significativity of the results and the differences among the different bands. It is necessary to insert densitometric analyses (Figg. 1d, 3a, 4a and 4c).
Figure 1e. The time-dependent effects are not as evident as the difference between gamma-ray and neutron treatment.
Paragraph 2.2: The results presented in Fig. 2 are not adequate to sustain the conclusion that “autophagy promotes cell death and contributes to cellular sensitivity”. They only describe the involvement of autophagy but not if the process represents a survival or cell death program.
The sentence (page 6) “Moreover, exposure to neutrons….” is ridundant.
Minor comments
Figure 1a-d: “at the indicated time points”: It is not clear what time points.
Figure 1c: It is necessary in Methods section to specify the difference between OS-26 and OS-32.
Figure 2c: To specify the difference among up and down panels.
It is often repeated that a result is “obvious”. But this is not always so.
Page 6: To correct p-Akt proteins in protein
Page 8: “Additionally, p.eIF2, p-JNK… showed a higher apoptosis rate”. It is not clear, perhaps “autophagy rate”.
Author Response
The article by Oh et al. explores the role of autophagy induced by high-linear energy transfer neutron in osteosarcoma models. The manuscript is well written, the aims and objectives are focused. However, many conclusions are not sustained by the experiments carried. Experimental plain, indeed, is sequential but the interpretation of the results appears to be very personal.
The supporting bibliography is ancient. From an analysis on PubMed it is possible to refer to much more recent articles, also leaving the works that are milestones of the topic.
Major comments
To sustain the conclusions about the role exerted by autophagy in the experimental model, it is necessary to carry out some experiments in which autophagy is inhibited either by specific chemical inhibitors or by silencing some protein markers. This will allow to better understand the role of the autophagic pathway in the experimental model studied.
Western blotting analyses do not report densitometric analysis. Therefore, it is not easy to analyse the significativity of the results and the differences among the different bands. It is necessary to insert densitometric analyses (Fig. 1d, 3a, 4a and 4c).
[Answer] We added data by quantifying the bands of western blotting.
Figure 1e. The time-dependent effects are not as evident as the difference between gamma-ray and neutron treatment.
[Answer] Our purpose was to show that gamma-ray and neutrons effectively suppressed the growth of cancer cells. Neutron, especially, more effectively inhibited cell growth than gamma rays. Figure 1e showed that neutrons inhibit cancer cell growth more than gamma rays at each time point. In view of your concerns, we have revised the sentence to “at each time point”.
Paragraph 2.2: The results presented in Fig. 2 are not adequate to sustain the conclusion that “autophagy promotes cell death and contributes to cellular sensitivity”. They only describe the involvement of autophagy but not if the process represents a survival or cell death program.
[Answer] Neutron attenuated the AKT/mTOR axis and increased p-JNK and p-eIF2α. mTOR is one of the central regulators of autophagy induction, and AKT modulates mTOR and regulates cell survival and proliferation [1]. Phosphorylation of eIF2α and JNK is known to be involved in autophagy induction [2]. During revision, we examined the effect of neutrons on autophagy and cell death using ly294002. We performed cell proliferation and cell counting, and WB to determine the involvement of autophagy and cell death by neutrons (fig2 e-g). We analyzed the involvement of autophagy and cell death by neutrons through the AKT/mTOR axis and JNK activation.
[1] Fan, S.; Zhang, B.; Luan, P.; Gu, B.; Wan, Q.; Huang, X.; Liao, W.; Liu, J., PI3K/AKT/mTOR/p70S6K Pathway Is Involved in Abeta25-35-Induced Autophagy. Biomed Res Int 2015, 2015, 161020
[2] heng, X.; Liu, H.; Jiang, C. C.; Fang, L.; Chen, C.; Zhang, X. D.; Jiang, Z. W., Connecting endoplasmic reticulum stress to autophagy through IRE1/JNK/beclin-1 in breast cancer cells. Int J Mol Med 2014, 34, (3), 772-81.
The sentence (page 6) “Moreover, exposure to neutrons….” is ridundant.
[Answer] We modified the expression.
Minor comments
Figure 1a-d: “at the indicated time points”: It is not clear what time points.
[Answer] Apoptosis (Figure 1b) were assessed after 72-h gamma-rays and neutrons and others were measured 48h later. Each time point was indicated in the legend section.
Figure 1c: It is necessary in Methods section to specify the difference between OS-26 and OS-32.
[Answer] We described it in the Method section. (See attached PDF file)
Figure 2c: To specify the difference among up and down panels.
[Answer] The below picture is a magnified version of the above picture to show autosome.
It is often repeated that a result is “obvious”. But this is not always so.
[Answer] We revised it as your request.
Page 6: To correct p-Akt proteins in protein
[Answer] We revised it as your request.
Page 8: “Additionally, p.eIF2, p-JNK… showed a higher apoptosis rate”. It is not clear, perhaps “autophagy rate”.
[Answer] We revised it as your request.

Reviewer 2 Report
This manuscript describes the behaviour of autophagy in radiosensitivity, cell survival, and cellular resistance against high-LET neutron radiation. Overall, the study is sound and in vivo murine cancer model support the data. Some technical issues here below are listed to improve its quality.
Fig 1d. Treatment with inhibitor of lysosomal degradation is required to determine whether the increase of LC3-II levels is due to a higher production of autophagosomes or to an inhibition of their degradation within lysosomes.
Fig 1e. Authors state 'Neutron treatment significantly inhibited cell growth, as determined by a trypan blue-exclusion assay in the two OS cell lines and primary cells derived from a patient with OS in a time-dependent manner (Fig. 1e).' To assess cell growth, cell cycle probes are more appropriate than tryptan blue.
Fig 3a and 4C. Please show blots of total proteins without phoshorylation (S6K, AKT and mTOR etc) to make sure the increase of the phosphorylation is not due to an upregulation of the level of the protein itself
Fig 5a. A quantification of limb radiographs would be insightful to assess convincingly tumour progression.
Author Response
Reviewer #2: REVIEWER COMMENTS
This manuscript describes the behaviour of autophagy in radiosensitivity, cell survival, and cellular resistance against high-LET neutron radiation. Overall, the study is sound and in vivo murine cancer model support the data. Some technical issues here below are listed to improve its quality.
Fig 1d. Treatment with inhibitor of lysosomal degradation is required to determine whether the increase of LC3-II levels is due to a higher production of autophagosomes or to an inhibition of their degradation within lysosomes.
[Answer] To clarify whether the increased LC3-II levels by neutrons are due to a higher production of autophagosomes or an inhibition of their degradation within lysosomes, we used CQ (blocking reagent of autophagosome formation) to determine the effect neutrons have on autophagy. Our results and descriptions were indicated in fig1.d.
Fig 1e. Authors state 'Neutron treatment significantly inhibited cell growth, as determined by a trypan blue-exclusion assay in the two OS cell lines and primary cells derived from a patient with OS in a time-dependent manner (Fig. 1e).' To assess cell growth, cell cycle probes are more appropriate than tryptan blue.
[Answer] We have added the data of the cell cycle using FACS analysis as per your request
Fig 3a and 4C. Please show blots of total proteins without phoshorylation (S6K, AKT and mTOR etc) to make sure the increase of the phosphorylation is not due to an upregulation of the level of the protein itself.
[Answer] We have added the data as your request.
Fig 5a. A quantification of limb radiographs would be insightful to assess convincingly tumour progression.
[Answer] We have added the data as your request.
Round 2
Reviewer 1 Report
The Authors responded adequately to the requests. The manuscript can be accepted for publication.